# Mice Generated with Induced Pluripotent Stem Cells Derived from Mucosal-Associated Invariant T Cells

**DOI:** 10.3390/biomedicines12010137

**Published:** 2024-01-09

**Authors:** Chie Sugimoto, Hiroyoshi Fujita, Hiroshi Wakao

**Affiliations:** Host Defense Division, Research Centre for Advanced Medical Science, Dokkyo Medical University, Mibu 321-0293, Japan; csugimot@dokkyomed.ac.jp (C.S.);

**Keywords:** MAIT cells, iPS cells, chimeric mice, rearranged TCR, anti-metastatic activity

## Abstract

The function of mucosal-associated invariant T (MAIT) cells, a burgeoning member of innate-like T cells abundant in humans and implicated in many diseases, remains obscure. To explore this, mice with a rearranged T cell receptor (TCR) α or β locus, specific for MAIT cells, were generated via induced pluripotent stem cells derived from MAIT cells and were designated Vα19 and Vβ8 mice, respectively. Both groups of mice expressed large numbers of MAIT cells. The MAIT cells from these mice were activated by cytokines and an agonist to produce IFN-γ and IL-17. While Vβ8 mice showed resistance in a cancer metastasis model, Vα19 mice did not. Adoptive transfer of MAIT cells from the latter into the control mice, however, recapitulated the resistance. These mice present an implication for understanding the role of MAIT cells in health and disease and in developing treatments for the plethora of diseases in which MAIT cells are implicated.

## 1. Introduction

MAIT cells are an emerging member of innate-like T cells. They are characterized by the semi-invariant T cell antigen receptor (TCR), dependence on major histocompatibility complex (MHC) I-related gene protein (MR1), on microbiota for their development, promyelocytic leukemia zinc finger protein (PLZF) transcription factor expression, and the retinoic acid receptor-related orphan receptor gamma t (RORγt) [1,2,3]. PLZF is associated with tissue homing and residency, conferring rapid T cell antigen responses in a TCR-independent manner, thus being essential for MAIT cell development and function [4,5]. RORγt is responsible for T helper (Th)-17 cell differentiation and for the development of type 3 innate lymphoid cells (ILC3) [6]. ILC is a collective term for lymphoid cells devoid of rearranged TCR and is composed of three members, ILC1, ILC2, and ILC3. Each member expresses a specific transcription factor responsible for the production of a particular cytokine(s). ILC1 is characterized by expression of the transcription factor T-bet, responsible for the production of IFN-γ, while ILC2 expresses the transcription factor GATA3 for IL-5 and IL-13. ILC3 produces IL-17 and IL-22 which are dependent on RORγt and the aryl hydrocarbon receptor (AHR), respectively [7,8]. Moreover, MAIT cells recognize small compounds, such as vitamin B2 metabolites, B9, and derivatives thereof, presented on MR1 via the semi-invariant T cell receptor (TCR) α (TRAV1-2-TRAJ33 (-TRAJ12, -TRAJ20)) in humans and TRAV1-TRAJ33 in mice, paired with a limited TCR β repertoire, as is the case for other innate-like T cells [9,10,11]. Whilst T cells, in adaptive immunity, necessitate priming and clonal expansion to exert their effector functions, innate-like T cells do not. This is due to the abundance of the latter, which are poised to respond to stimuli in vivo. MAIT cells are implicated in an array of diseases, such as bacterial and viral disease, autoimmune, inflammatory, and metabolic diseases, asthma, and cancer [2,12,13]. However, the precise role of MAIT cells in each disease is enigmatic despite their abundance in humans, representing 1–10% of T cells in peripheral blood mononuclear cells (PBMC) and 20–45% in the liver [14]. Mouse studies are an effective approach to elucidating the function of MAIT cells in health and disease. Nonetheless, mouse MAIT cells are rare, representing 1/10~1/1000 of human cells, making it difficult to study their functions [12]. Although several MAIT cell-rich mice have been reported, intrinsic problems remain, such as the aberrant expression of the transcription factor PLZF in MAIT-TCR transgenic mice and the Cast origin of *Trav1-Traj33* in Cast-B6 mice, which may limit the use of MAIT cells in a strict immunological setting [15,16]. The higher frequency of MAIT cells in Cast-B6 mice may make them suitable in elucidating the function of MAIT cells in immunity, but there are few data on the advantage of Cast-B6 mice over wild-type mouse strains such as C57BL/6 and BALB/c. Therefore, to explore the immunological functions of MAIT cells, it is necessary to overcome the above problems and create new strains of mice with increased MAIT cells.

Our recent study established induced pluripotent stem cells (iPSCs) from murine MAIT cells (MAIT-iPSCs). MAIT-iPSCs exclusively gave rise to MAIT-like cells as defined 5-(2-oxopropylideneamino)-6-D-ribitylaminouracil (5-OP-RU)-loaded mouse MR1-tetramer (mMR1-tet)^+^TCRβ^+^ cells, when differentiated under T cell permissive culture conditions. These MAIT-like cells shed light on possible MAIT cell roles in cancer immunology [17]. Previously, we established cloned mice and cloned embryonic stem (ntES) cells by nuclear transfer of iNKT cells, another member of the innate-like T cell group [18,19]. The cloned mouse offspring with rearranged TCRα specific for iΝΚΤ cells (*Trav11-Traj18*) showed an enhanced frequency of iNKT cells [20], and the ntES cells exclusively redifferentiated into iNKT-like cells in the OP9 and OP9/delta-like 1 (DLL1) system [20]. Therefore, we reasoned that mice with rearranged TCRα and/or β specific for MAIT cells will have an increased frequency of MAIT cells. Given that mouse iPSCs possess pluripotency, including the ability to give rise to chimeric mice as ES cells, we explored the ability of MAIT-iPSCs to confer the rearranged TCR loci to the mice [21].

In this study, mice were generated from MAIT-iPSCs with a rearranged TCRα or β locus, specific for MAIT cell TCR, designated as Vα19 and Vβ8 mice, respectively. TCR repertoire analysis on *Trav* in Vβ8 mice revealed greater diversity in the CDR3α domain than anticipated. While both mice possessed increased MAIT cell numbers, Vβ8 mice exhibited resistance to inoculated cancer, but Vα19 mice did not. Intriguingly, the adoptive transfer of MAIT cells from Vα19 mice into the control mice conferred a similar resistance. Further study identified T cells, particularly CD8 T cells, as an adjuvant for MAIT cells to exert anti-tumor immune functions.

## 2. Materials and Methods

### 2.1. Mice 

Mice were housed in the Animal Research Center, Dokkyo Medical University, under specific pathogen-free conditions with controlled lighting and temperature with food and water provided ad libitum. For breeding, male and female mice aged between 8 to 30 weeks were used. For tissue cell analyses and cancer models, male and female mice aged between 8 to 12 weeks were used.

### 2.2. Cell Lines

Mouse cancer cell line Lewis lung carcinoma (LLC) was cultured in DMEM supplemented with 10% FBS at 37 °C in 5% CO_2_. 

### 2.3. Generation of Chimeric Mice from MAIT-iPSCs

iPSCs were established previously from lung MAIT cells of male C57BL/6NJcl mice [17]. MAIT-iPSC clones L7-1, L11-1, and L19-1 were selected by karyotype analysis (80–90% euploid metaphases) and microinjected into ICR 8-cell-stage embryos at NPO for Biotechnology Research and Development, Osaka University (Osaka, Japan) (Appendix A). 

### 2.4. Generation of Vα19 and Vβ8 Mice from the Chimeric Mouse

To confirm germline transmission of an allele derived from MAIT-iPSCs, the male chimeras were crossed with C57BL/6NJcl females (CLEA, Tokyo, Japan) and progenies with black coat color were genotyped for the rearranged TRA (*Trav1-Traj33*) and TRB (*Trbv13-3-d1-j1-2* or *Trbv19-1-d2-j2-3)* loci for MAIT cells by PCR (Appendix A). Offspring of a germline-transmitted chimera (#26) harboring *Trav1-Traj33* or *Trbv13-3-d1-j1-2* in an allele were crossed to obtain homozygous mice for the rearranged locus, and the obtained mice were further crossed with C57BL/6NJcl (CLEA, Tokyo, Japan) to generate hemizygous mice harboring either *Trav1-Traj33* (Vα19 mice) or *Trbv13-3-d1-j1-2* (Vβ8 mice). Vα19/Vβ8 mice were hemizygous for both *Trav1-Traj33* and *Trbv13-3-d1-j1-2*.

### 2.5. Mouse Genotyping

Ear punches from mice were treated in 50 mM NaOH at 95 °C for 10 min and centrifuged at 15,000 rpm for 10 min. The supernatant was used as a genomic DNA sample after neutralization. Primers were designed to detect the rearranged configuration of TRA and TRB specific for MAIT-iPSC using Primer-BLAST (Appendix A). PCR amplification was performed with specific primers and OneTaq master mix (New England Biolabs, Tokyo, Japan) under the following conditions: 94 °C for 30 s, 35 cycles of 94 °C for 30 s, 55 °C for 30 s, and 68 °C for 30 s, and 68 °C for 3 min. PCR products were analyzed on 2% agarose gel electrophoresis (Appendix A).

### 2.6. Cell Isolation from Tissues

Mononuclear cells were isolated from the spleen, thymus, lymph nodes, lung, liver, and intestine as described previously [17]. 

### 2.7. Flow Cytometry 

Cells were stained with the antibodies listed in the key resource table. For analyses of cell surface markers, 7-AAD was used to discriminate between live and dead cells prior to staining, and samples with cell viability greater than 97% were used for staining. For expression analysis of the transcription factors in MAIT cells, the cells were stained with anti-TCRβ and mMR1-tet then fixed and permeabilized with a Transcription Factor buffer set (BD Biosciences, Tokyo, Japan) according to the manufacturer’s instruction. Then, the cells were stained with anti-PLZF, -T-bet, and -RORγt antibodies. For intracellular cytokine staining, the cells were treated with a different combination of the cytokines, Cell Stimulation Cocktail (Thermo Fisher Scientific, Tokyo, Japan), or 5-OP-RU for 15 h. Protein Transport Inhibitor Cocktail (ThermoFisher Scientific, Tokyo, Japan) was added 1.5 h after the start of stimulation. Thereafter, the cells were subjected to intracellular staining with a BD Cytofix/Cytoperm buffer kit according to the manufacturer’s instruction (BD BioScience, Tokyo, Japan). Cells were analyzed with the MACSQuant cell analyzer (3 lasers, 10 parameters, Miltenyi Biotec, Tokyo, Japan) or the AttuneNxT acoustic focusing cytometer (4 lasers, 14 parameters, Thermo Fisher Scientific, Tokyo, Japan). Data were processed using FlowJo software (ver. 9 or 10, BD Biosciences, Tokyo, Japan). Cell sorting was performed using a FACSJazz cell sorter (2 lasers, 8 parameters, BD Biosciences, Tokyo, Japan). 

### 2.8. MAIT Cell Activation

Cells from lungs (5 × 10^5^ cells/well) or spleens (1 × 10^6^ cells/well) of Vα19 and Vβ8 mice were stimulated with cytokines (200 ng/mL each) or 5-OP-RU (1–100 nM) in 96-well culture plates at 37 °C, 5% CO_2_ for 18 h. The culture supernatants were collected for the cytokine assay described below and the cells were stained to determine MAIT cell activation by flow cytometry. 

### 2.9. TCR Repertoire Analysis

MAIT cells (mMR1-tet^+^TCRβ^+^ cells) and non-MAIT T cells (mMR1-tet^-^CD4^+^CD8^+^TCRβ^+^ cells) were sort-purified from thymocytes of Vα19 and Vβ8 mice and stored into RNAlater (Qiagen, Tokyo, Japan). Total RNA was isolated using an RNeasy Mini kit (Qiagen, Tokyo, Japan) and resultant RNAs (10 ng per sample) were subjected to cDNA library construction (SMARTer Mouse TCR a/b profiling kit, Takara Bio, Shiga, Japan). Next-generation sequencing was performed with MiSeq (Illumina, CA, USA) at FASMAC (Kanagawa, Japan). Sequencing data were aligned to reference mouse TCR V/D/J sequences registered in the ImMunoGeneTics database and then assembled into TCR clones using MiXCR-3.0.5 [22] and TCR repertoire data were visualized using VDJtools-1.2.1 [23] at ImmunoGeneTeqs, Inc. (Chiba, Japan).

### 2.10. Tumor Resistance 

LLC suspended in HBSS was intravenously (I.V.) inoculated (3.0 × 10^5^ cells/mouse) into the indicated mouse strains. For MAIT cell supplementation experiments, MAIT cells were isolated from lungs and spleens of Vα19 mice. In brief, cells were stained with APC-labeled mMR1-tet and mMR1-tet-positive cells were isolated with anti-APC microbeads and LS columns (Miltenyi Biotec, Tokyo, Japan). The isolated MAIT cells were intraperitoneally (IP) injected (1.0 × 10^6^ cells/mouse) into control (C57BL/6) mice 3 days before LLC inoculation. For CD3^+^ T cell and CD8^+^ T cell supplement experiments, CD3^+^ T cells were purified from C57BL/6 spleen cells with a MojoSort mouse CD3 T cell isolation kit (Biolegend, CA, USA), while CD8^+^ T cells were isolated by adding biotin-labeled anti-CD4 antibody into a MojoSort mouse CD3 T cell isolation kit. The purified cells were IP injected (3.0 × 10^5^ cells/mouse) into Vα19 mice 3 days before LLC inoculation. In the survival assay, mice were considered to be dead when they showed a humane endpoint, such as acute weight loss, hypothermia, or severe gait and/or consciousness disturbance. A Kaplan–Meier survival plot was calculated based on the survival time of mice and the percentage of survived mice at the indicated point.

### 2.11. Quantification and Statistical Analysis

Statistical analyses were conducted using Prism 9.9.6 for macOS (GraphPad, Boston, MA, USA). In each experiment, data variability was shown as the standard deviation (SD). One-way ANOVA was used for analysis of the activation by cytokines, of the intracellular cytokines, and of the effector molecule production. The log-rank test was used for survival analyses between the two indicated groups.

### 2.12. Relevant Materials Used in this Study

Relevant materials used in this study are summarized in Appendix B.

## 3. Results

### 3.1. Mice Harboring Rearranged TCR Loci Specific for MAIT Cells 

We previously established several iPSC clones derived from 5-OP-RU-loaded mouse MR1 tetramer (mMR1-tet)-positive MAIT cells and confirmed that some clones possessed pluripotency through chimera formation [17]. Using these chimeric mice, the generation of novel C57BL/6 strains harboring rearranged TCR loci specific for MAIT cells was attempted. Three MAIT-iPSC clones, namely L7-1, L11-1, and L19-1, were injected independently into ICR 8-cell-stage embryos, resulting in 11 chimeric male mice with varying degrees of chimerism (<30−90% as defined by donor coat color) (Table 1 and Appendix A). Mice with 60~90% chimerism were crossed with C57BL/6 females and the resulting black-colored pups were screened for the rearranged configuration of *Trav1-Traj33* and/or *Trbv* corresponding to the original iPSC clones (Appendix A–D). Among the chimeric mice, one line with 60–90% chimerism (#26 derived from iPSC clone L7-1 harboring *Trav1-Traj33* and *Trbv13-3-d1-j1-2* [17]) successively gave rise to offspring bearing *Trav1-Traj33*, *Trbv13-3-d1-j1-2*, or both (Table 1, Appendix A–E). Hereafter, these mouse strains are designated Vα19 (harboring *Trav1-Traj33*), Vβ8 (harboring *Trbv13-3-d1-j1-2*), and Vα19/Vβ8 (harboring both *Trav1-Traj33* and *Trbv13-3-d1-j1-2*), respectively. 

To visualize the impact of the rearranged TCRα or TCRβ locus specific for MAIT cells on the frequency of MAIT cells, flow cytometric analysis was performed using peripheral blood (Figure 1). While the frequency of MAIT cells (defined as mMR1-tet^+^TCRβ^+^ cells) in the control mice rarely exceeded 0.1% among αβ T cells, in Vβ8 mice and Vα19 mice, it reached ~0.5% and ~36%, respectively (Figure 1A,B). Furthermore, the presence of *Trav1-Traj33* and *Trbv13-3-d1-j1-2* enhanced the frequency (Figure 1B). CD4 and CD8 expression analysis identified an increase in CD4^−^CD8^−^ MAIT cells concomitant with a decrease in CD4^+^ MAIT cells, relative to the control mice (Figure 1C). In contrast, among mMR1-tet^−^TCRβ^+^ cells (non-MAIT T cells) in Vβ8 mice, there was little change in the frequency of CD4^+^ and CD8^+^ as well as CD4^−^CD8^−^ T cells. However, relative to the control, the frequency of CD4^−^CD8^−^ T cells in Vα19 mice increased, and, in Vα19/Vβ8 mice, CD8^+^ T cell frequency decreased (Figure 1D). Hereafter, all experiments are performed with Vα19 and Vβ8 mice since the presence of both in-frame rearranged *Trav1-Traj33* and *Trbv13-3-d1-j1-2* as seen in Vα19/Vβ8 mice may limit the repertoire and function of MAIT cells.

PLZF, RORγt, and T-bet are pivotal transcription factors in MAIT cell development and function [5,15,16]. Thus, their expression in MAIT cells from the spleen and lungs of Vα19 and Vβ8 mice was examined. PLZF was detected in the majority of spleen and lung MAIT cells for both Vα19 and Vβ8 mice (Figure 1E). RORγt was also detected in MAIT cells from both groups of mice. However, lung MAIT cells exhibited higher expression than spleen MAIT cells (Figure 1E). MAIT cells are composed of the MAIT1 and MAIT17 subsets; however, their distribution varies from one tissue to another in C57BL/6 [24]. MAIT17 is characterized by RORγt expression, whereas MAIT1 is characterized by T-bet expression. The relative abundance of MAIT1 and MAIT17 was then evaluated in these mice. In both Vα19 and Vβ8 mouse lung MAIT cells, RORγt-expressing cells (MAIT17) were dominant over T-bet-expressing cells (MAIT1) (~60–80% cells were RORγt^+^, while less than 20% cells were T-bet^+^). In contrast, MAIT1 was more abundant in spleen MAIT cells (40–60% of cells were T-bet^+^, while less than 10% and 40% were RORγt^+^ in Vα19 mice and Vβ8 mice, respectively (Figure 1F)). These results agree with the previous report evaluating C57BL/6 [24].

Given the elevated frequency of MAIT cells in the blood, whether Vα19 and Vβ8 mice also harbored an enhanced MAIT cell frequency across the organs was evaluated (Figure 2). In Vα19 mice, MAIT cells constituted ~20−25% of CD3^+^ T cells in the spleen, inguinal and mesenteric lymph nodes, intestinal lamina propria (LP), and the thymocytes, while increased MAIT cell numbers were observed in the liver, lung, and bone marrow (~35−40% among CD3^+^ T cells) (Figure 2B). In Vβ8 mice, albeit to a lesser extent than Vα19 mice, MAIT cells represented 2−10% of CD3^+^ T cells in the analyzed tissues (Figure 2B). Finally, the control mice exhibited a much lower MAIT cell frequency. 

As *Trav1* is located at the most extreme in the *Trav* locus, rearranged *Trav1*-*Traj33* is unlikely to undergo further rearrangement (secondary rearrangement) in the same allele. Thus, whether *Trav1*-*Traj33* impacted the generation of other T cells such as innate-like T cells and conventional T cells in adaptive immunity was evaluated (Figure 2C–F). The frequency of iNKT cells, the most abundant mouse innate-like T cell subset, was reduced in the liver of Vα19 mice relative to control and Vβ8 mice (Figure 2C). In sharp contrast, the frequency of γδT cells decreased in Vβ8 mice, but Vα19 showed little difference compared with control mice (Figure 2D). While the frequency of CD8^+^ T cells among Vα19, Vβ8, and control mice did not differ across the tissues, except for a decrease in the bone marrow and the lung of Vα19 mice, that of CD4^+^ T cells in Vα19 mice tended to decline in all tissues examined relative to other mouse strains (Figure 2E,F). 

Analysis of myeloid and lymphoid cells identified a decrease in T cell frequency in the intestinal LP and lung lymphocytes among CD45^+^ cells in Vα19 mice relative to the other strains. Nonetheless, the frequency of B cells (CD45^+^CD11b^−^CD3^−^CD19^+^ cells), NK cells (CD45^+^CD11b^−^CD3^−^CD19^−^NK1.1^+^NKG2D^+^ cells), macrophages (CD45^+^ CD11b^dim^), neutrophils (CD45^+^CD11b^+^Ly6G^+^ cells), and monocytes (CD45^+^CD11b^+^Ly6C^+^ cells) was unaffected in both Vα19 and Vβ8 mice (Appendix A). 

In summary, the data demonstrated that the rearranged TCR configuration specific for MAIT cells affected T cell development and enhanced the frequency of MAIT cells in mice. 

### 3.2. Expression of Molecules Relevant to MAIT Cells

Comparative analysis of MAIT and non-MAIT T cells in the representative tissues of Vα19 and Vβ8 mice was performed based on the cell surface molecules [15,16]. In Vα19 mice, little difference in surface molecule expression was observed between MAIT and non-MAIT T cells for any molecules except for NK1.1 in the spleen and in the intestinal LP, while there was an increase in interleukin (IL)-18Rα and CXCR6 in the liver and in lung MAIT cells relative to non-MAIT T cells (Figure 3). In contrast, between MAIT and non-MAIT T cells of Vβ8 mice, there was a large difference in surface molecule expression levels (Figure 3). MAIT cells in Vβ8 mice expressed higher levels of the surface molecules except for CD62L, which was expressed less than non-MAIT T cells across the tissues, except for intestinal LP. CD103 showed a similar decrease in MAIT cells in intestinal LP, whilst the reverse trend was seen in the spleen, liver, and lung in Vβ8 mice (Figure 3). 

### 3.3. TCR Repertoire of Vα19 and Vβ8 Mice 

Though the rearranged *Trav19-Traj33* and/or *Trbv13-3-d1-j1-2* enhanced the frequency of MAIT cells, the impact of such gene rearrangement on the TCR repertoire in the thymus was unknown. Thus, the TCR repertoire in Vα19 and Vβ8 mouse thymi was analyzed. MAIT cells (mMR1-tet^+^TCRβ^+^ cells) and non-MAIT T cells (mMR1-tet^−^CD4^+^CD8^+^TCRβ^+^) were sorted from the thymocytes and subject to high-throughput sequencing. While TCRα chains in MAIT cells were mostly composed of *Trav1-Traj3*3 in both mouse strains (94.6% in Vα19 mice and 85.1% in Vβ8 mice), other combinations such as *Trav16D-Traj18* (2.2%), *Trav1-Traj30* (0.3%), *Trav1-Traj31* (0.3%), *Trav1-Traj27* (0.1%), and *Trav1-Traj12* (0.1%) were also found in Vβ8 mice (Figure 4A and Appendix A) [25,26]. Further analysis of *Trav1-Traj3*3 revealed that TCRα complementarity determining region 3 (CDR3α) sequences were quasi-invariable in Vα19 mice, whereas in Vβ8 mice, they were highly divergent both in length and in sequence (Figure 4C) [16]. 

TRBV repertoire usage was quasi-exclusively limited to *Trbv13-3-d1-Trbj1-2* in Vβ8 mouse MAIT cells. In contrast, no such exclusive usage was observed in the TRBV repertoire of Vα19 mice, although there existed biased use of *Trbv5, 13-3, 13-2*, and *19* (Figure 4A). As expected, CDR3β sequences of *Trbv13-3-d1-Trbj1-2* in Vβ8 mouse MAIT cells exclusively comprised a single clone, while those for Vα19 mouse MAIT cells were divergent (Appendix A).

The TRAV repertoire of non-MAIT T cells in Vα19 mice showed enrichment in *Trav1-Traj3*3 similar to MAIT cells, while that in Vβ8 mice represented much more diversity. Regardless of *Trav1-Traj3*3 enrichment in non-MAIT T cells, the TRBV repertoire was less biased in Vα19 mice. In sharp contrast, *Trbv13-3-d1-Trbj1-2* was always predominant in Vβ8 mice, even in non-MAIT T cells (Figure 4B) [16]. 

### 3.4. MAIT Cell Activation and Production of IFN-γ and IL-17A

MAIT cells are activated to produce cytokines such as IFN-γ and IL-17A in a TCR-dependent and/or independent manner [16,27,28,29]. Our recent study also found that redifferentiated MAIT cells from MAIT-iPSCs in vitro possessed similar properties [17]. Therefore, cells isolated from the spleen and the lung of Vα19 and Vβ8 mice were stimulated with varying concentrations of the MAIT cell agonist 5-OP-RU or cytokines, and the resulting activation through (or independent of) TCR was evaluated by CD69 and CD25 expression on MAIT cells (Figure 5A–C and Appendix A, Appendix A). While activation of the lung and spleen MAIT cells from Vα19 mice was observed in a 5-OP-RU dose-dependent manner, that from Vβ8 mice plateaued at 10 nM, resulting in less enhanced activation (Figure 5A). Notably, Vβ8 mouse lung MAIT cells already showed a degree of CD25 and CD69 expression in the absence of stimulation (Figure 5A).

Subsequently, TCR-independent activation of MAIT cells from the lung and spleen of Vα19 mice was assessed with cytokines such as IL-12, IL-18, IL-15, IL-33, and TL1A [28]. Stimulation with IL-12 or IL-18 alone resulted in little activation, while the combination of cytokines resulted in activation, which was further enhanced by IL-15 or TL1A addition (Figure 5B and Appendix A). However, IL-33 failed to induce activation or enhance other cytokine-induced activation in Vα19 mice (Figure 5B and Appendix A). In contrast, Vβ8 mouse lung MAIT cells, exhibiting an activated phenotype without stimulation, were further activated when stimulated with IL-12 or IL-18 alone (Figure 5C). The combination of IL-12/IL18 or IL-12/IL-18/IL-15 led to the highest activation, but the addition of TL1A was not effective in boosting activation as seen in MAIT cells from Vα19 mice (Figure 5C and Appendix A). While activation of Vβ8 mouse spleen MAIT cells fluctuated among individual samples when stimulated with a single cytokine, it was minimized when stimulated with multiple cytokines (Figure 5C) [27,28,30]. 

Then, we addressed whether such activation resulted in the production of cytokines such as IFN-γ and IL-17A from MAIT cells by intracellular staining. This revealed that ~10% of MAIT cells in Vα19 and Vβ8 mouse spleens produced IFN-γ upon PMA/ionomycin or IL-12+IL-18+IL-15+ TL1A, while ~6% of spleen MAIT cells produced IL-17A upon PMA/ionomycin (Figure 5D,E). Similarly, PMA/ionomycin enhanced the frequency of granzyme B (an effector molecule required for cytotoxic activity)-expressing MAIT cells in Vα19 and Vβ8 mouse spleens, while perforin expression was not enhanced (Figure 5F,G). The data indicate that MAIT cells could produce IFN-γ, IL-17A, and granzyme B in Vα19 and Vβ8 mice.

### 3.5. Tumor Resistance in Vβ8 and Vα19 Mice

Our previous study demonstrated that MAIT-like cells prepared from iPSCs exert anti-tumor activity, increasing mouse survival upon adoptive transfer [17]. Therefore, whether the Vα19 and Vβ8 mice exhibited a similar tumor resistance was explored. Intravenous inoculation of Lewis lung carcinoma (LLC) into Vβ8 mice led to an increased mouse survival time relative to control mice, an effect not observed in Vα19 mice (Figure 6A). It is possible that MAIT cells in Vα19 mice were defective with respect to anti-tumor activity. Therefore, whether the adoptive transfer of MAIT cells from Vα19 mice into wild-type mice improved survival for the LLC-inoculated mice was examined, which it did. This indicated that the intrinsic anti-tumor activity of MAIT cells is preserved in Vα19 mice (Figure 6B). As the T cell repertoire of Vα19 mice was reduced at the cost of MAIT cell increase (Figure 4 and Appendix A), whether supplementing with wild-type mouse T cells improved the survival of LLC-inoculated Vα19 mice was examined. Given that T cells, as represented by cytotoxic CD8^+^ T cells, are pivotal for antitumor activity, CD3^+^ T cells or CD8^+^ T cells were adoptively transferred into Vα19 mice. The survival curves upon LLC inoculation for control and Vα19 mice were similar; however, Vα19 mice with the adoptive transfer of CD3^+^ T cells or CD8^+^ T cells from the control mice experienced increased survival time (Figure 6C). These data indicate that other subsets of immune cells are required for MAIT cells to exert effective anti-tumor activity. 

## 4. Discussion

In this study, novel mice models were created to facilitate the functional analysis of MAIT cells. Previous mouse models for studying the functions of MAIT cells were limited to MR1 knockout (KO) and/or MAIT TCR transgenic mice [31,32,33,34]. While the former allows study on mice devoid of MAIT cells, such mice also lack other MR1-restricted immune cells, making physiological changes difficult to ascribe to MAIT cells alone. Moreover, although data from MR1 KO mice are generally compared to those from wild-type mice, data interpretation is often complicated and unconvincing as laboratory mice produce few MAIT cells. Transgenic mice overexpressing TCR specific for MAIT cells can overcome this inherent problem but still suffer from aberrant MAIT cell function and/or the absence of other T cells [2]. In this respect, it is interesting to note that CAST/EiJ, a strain expressing more MAIT cells than other laboratory strains, exhibited diminished *Mycobacteria tuberculosis* virulence [35]. Furthermore, Cast/B6 mice were generated by crossing CAST/EiJ with C57BL/6 to overcome the problem of MAIT cell paucity, although the effect on disease remains unknown [15]. It is noteworthy that mice lacking γδ T cells and/or iNKT cells show an increase in MAIT cells [36]. However, under normal circumstances, it is likely that there is competition between MAIT cells and these cells for the niche. Thus, the loss of γδ T cells and/or iNKT cells would provide a niche for MAIT cells to develop favorably. 

In addition to these mice, novel mice derived from iPSCs generated from lung MAIT cells are reported here. They were characterized by the presence of the rearranged *Tcra* or *Tcrb* locus in the allele, both of which increased MAIT cell expression when inherited separately. The presence of rearranged *Trav1-Traj33* in all cell alleles, including immature T cells such as MAIT cell precursors (which possess the Trav1-Traj33 transcript but have not yet been selected by MR1), favors biased selection of MAIT cells because there are many cells expressing the transcript during T cell development in Vα19 mice. Given that the selection of *Trav* and *Traj* DNA fragments is random during TCRα gene rearrangement and that only a small population of precursor cells harboring TRAV1-TRAJ33 are selected by MR1 in WT mice, this abundance of cells harboring the Trav1-Traj33 transcript would result in far more MAIT cells in Vα19 mice than in WT mice. In contrast, the number of MAIT cells is relatively low in Vβ8 mice. This is in part due to the absence of the rearranged *Trav1-Traj33* in the allele of Vβ8 mice. However, the presence of a rearranged *Trbv13-3-d1-j1-2* in the allele derived from the MAIT cell TCR would allow preferential and/or privileged pairing with nascent MAIT cells harboring TRAV1-TRAJ33. Such pairing is further enhanced by allelic exclusion imposed by *Trbv13-3-d1-j1-2*. These features result in an increase in MAIT cells in Vβ8 mice compared to WT mice. Notably, Vα19 and Vβ8 mice have abundant endogenous MAIT cells that persist throughout life. In contrast, reMAIT cells adoptively transferred into C57BL/6 did not [17]. Thus, the former mice would be useful for studying the long-term function of MAIT cells, such as in metabolic diseases, while the latter model may be suitable for assessing the relatively short- to medium-term function of MAIT cells, as represented by acute infections, cancer, and asthma.

Although the frequency of MAIT cells in Vα19 mice was higher than in Vβ8 mice, flow cytometric and TCR repertoire analyses revealed the presence of many other T cells including iNKT cells and γδT cells in Vα19 mice (Figure 2 and Figure 4). As *Trav1* is located at the most distal region of the TCRα locus, once MAIT cell-specific TCRα (*Trav1-Traj33*) was selected, iNKT cell-specific TCR (*Trav11-Traj18*) and/or any other *Trav-Traj* rearrangement could not occur in the same allele. Thus, it is plausible that V-J recombination from the other allele is at work in these mice to ensure TCRα chain diversity to a certain degree, thus allowing iNKT cell and γδ T cell development. 

While CDR3α in thymic MAIT cells from Vα19 mice showed a single sequence, Vβ8 mice exhibited sequence diversity and V-J combination was not confined to *Trav1-Traj33*, counter to previous studies (Figure 4C and Appendix A) [16,26]. This unexpected CDR3α diversity may reflect the presence of pre-MAIT cells in the thymus and the survival or selection of only a fraction before and/or after their egression from the thymus. Furthermore, the abundance of the Trav1-Traj33 transcript, a major combination for the MAIT TCRα chain, in mMR1-tet-negative T cells described as “non-MAIT T cells” from Vα19 mice may suggest that other MAIT cell subsets, unrecognized by 5-OP-RU-loaded mMR1-tet, were enriched in the thymus. Alternatively, such an increase may reflect the fact that 5-OP-RU-loaded mMR1-tet no longer recognized the MAIT-TCR, whose level was below the detection limit, due to the large accumulation of rearranged *Trav1-Traj33* in the cytoplasm, which interfered with and restricted translation.

As PLZF expression is essential for MAIT cell development and function, our data indicate that MAIT cells in Vα19 and Vβ8 mice are functional (Figure 1E). Furthermore, expression of RORγt and/or T-bet in MAIT cells concomitant with the production of IL-17A and/or IFN-γ further underpins the notion that MAIT cells in these mice are authentic and distinguished from those found in Vα19-Cα^−/−^ transgenic mice (Figure 1E,F and Figure 5D,E) [16]. 

While the distributions of MAIT1 and MAIT17 in Vα19 mice and Vβ8 mice are similar to that in C57BL/6, MAIT cells are yet to be analyzed by single-cell RNA sequencing across tissues. This may further support the above notion and shed light on tissue-specific MAIT cell subset distribution and function. 

The fact that MAIT cells in Vβ8 mice showed an activated phenotype and produced granzyme B, an effector molecule, in the absence of external stimuli ex vivo may indicate that MAIT cells are present as effector cells in vivo (Figure 5A,F and Appendix A). This may in part explain why 5-OP-RU only weakly activated Vβ8 MAIT cells ex vivo (Figure 5A).

Regardless of the increase in MAIT cells concomitant with T cells harboring V-J combinations other than *Trav1-Traj33* in Vα19 mice, it was insufficient to fully recover CD8^+^ T and CD4^+^ cell frequency. In particular, the decrease in bone marrow and lung CD8^+^ T cells for Vα19 mice may directly lead to compromised anti-metastatic activity (Figure 2E and Figure 6A). While our previous study demonstrated that NK cells are an important partner for MAIT cells to exert anti-metastatic function, this study further implied the requirement for CD8^+^ T cells [17]. This indicates that both NK and CD8^+^ T cell interaction with MAIT cells may be implicated in anti-metastatic activity. In this respect, it will be interesting to examine whether IL-2 and IL-15 have an effect on NKG2D and CD158a-b in MAIT cells, because the cytokines enhance NKG2D expression in different lymphocytes and appear to increase the anti-tumor potential of NK cells [37]. Moreover, the influence of CD8^+^ T cells on MAIT cells in tumor immunology warrants further study. Given that the frequency of CD4^+^ T cells was affected in Vα19 mice, future studies must examine the mechanism(s) by which *Trav1-Traj33* in the allele engendered the CD4^+^ T cell decrease. As MR1 belongs to MHC Ib, it is tempting to postulate competition between MHC II and MHC Ib but not between MHC I and MHC Ib in T cell development. 

### Limitations of the Study

Though, herein, mice are reported with increased MAIT cell numbers, equivalent to or superior to that of humans, without genetic modification and their utility in cancer immunology explored, not all MAIT cell functions in Vα19 and Vβ8 mice were investigated. In particular, the effects of allelic exclusion in the presence of the rearranged *Trbv* on MAIT cell activation, cytokine production, and cytolytic activity in Vβ8 mice are yet to be examined. MAIT cell self-activation was observed despite the absence of obvious exogenous stimuli in Vβ8 mice (Figure 5A,C and Appendix A). It is possible that allelic exclusion, imposed by rearranged TRBV, renders Vβ8 mouse highly susceptible to environmental cues (microbial, commensal, and superantigen stimulation). Alternatively, Vβ8 mice may promote MAIT cell maturation and activation more efficiently than B6 mice, provided that the lymphocyte niche is the same. As a result, MAIT cells in Vα19 mice did not show such activation as there were too many cells to efficiently promote maturation and activation. Further studies are needed, in both cases, to decipher the mechanisms underlying such activation.

Similarly, the nature of immune cells including MAIT cells in Vα19 mice requires further exploration. 

MAIT cells from Vα19 mice exhibited anti-metastatic properties upon adoptive transfer. However, it remains undetermined whether MAIT cells were composed of a unique population or of different cell subsets with diverse functions and different developmental stages. Moreover, tissue-specific MAIT cell characterization is fundamental for clarifying MAIT cell function in each tissue and in understanding the integral role of MAIT cells in health and disease.

## 5. Resource Availability 

### 5.1. Lead Contact 

Further information and requests for resources and reagents should be directed to and will be fulfilled by the Lead Contact, Hiroshi Wakao (hwakao@dokkyomed.ac.jp).

### 5.2. Materials Availability

The mouse strains (Vα19 and Vβ8) generated in this study will be deposited to a public biological resource bank and be made available to qualified investigators with a Material Transfer Agreement.

## Figures and Tables

**Figure 1 biomedicines-12-00137-f001:**
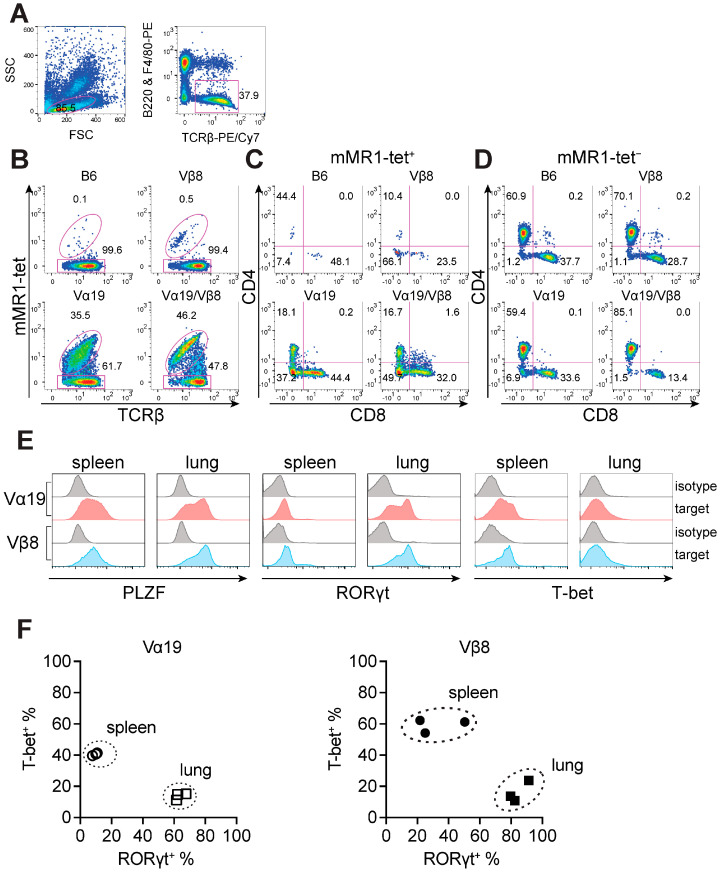
Blood MAIT cell profile in C57BL/6 (control), Vα19, Vβ8, and Vα19/Vβ8 mice. Anticoagulated mouse peripheral blood collected by submandibular bleeding was used for flow cytometric analysis. (**A**) Gating for T cell population (TCRβ^+^ lymphocytes). (**B**) Analysis of MAIT cells (mMR1-tet^+^TCRβ^+^) and non-MAIT T cells (mMR1-tet^=^TCRβ^+^) in each mouse strain. (**C**,**D**) The expression of CD4 and CD8 in mMR1-tet^+^ TCRβ^+^ cells (**C**) and mMR1-tet^=^TCRβ^+^ cells (**D**). B6: C57BL/6, Vβ8: Vβ8 mice, Vα19: Vα19 mice, Vα19/Vβ8: Vα19/Vβ8 mice. The number shows the percentage of the indicated cell subset. Representative data from three independent experiments are shown. (**A**–**D**) Pseudcolor density plots are shown. (**E**) Representative histograms showing the expression of PLZF, RORγt, and T-bet in Vα19 and Vβ8 mouse spleen and lung MAIT cells (shaded in light orange and in light blue, respectively). Isotype control staining is also indicated (shaded in grey). Representative data from three independent experiments are shown. (**F**) MAIT1 and MAIT17 in spleen and lung. Scatter plot showing the percentage of lung and spleen MAIT cells expressing T-bet (MAIT1) and/or RORγt (MAIT17) in Vα19 mice (left panel, Vα19) and Vβ8 mice (right panel, Vβ8) (*n* = 3 per strain).

**Figure 2 biomedicines-12-00137-f002:**
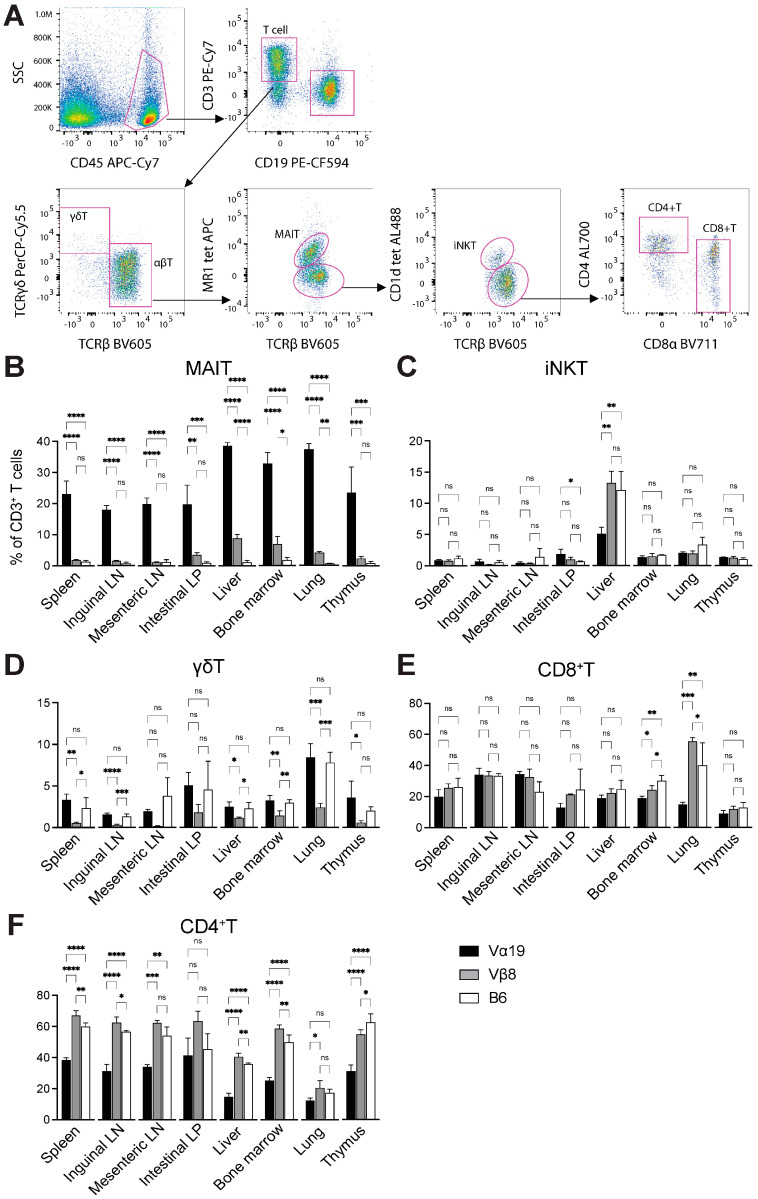
Frequency of T cell subsets in Vα19, Vβ8, and control mice. Representative gating strategy of each T cell subset (pseudcolor density plots) is shown in (**A**). Relative frequency of MAIT (**B**), iNKT (**C**), γδT (**D**), CD8^+^ T (**E**), and CD4^+^ T (**F**) cells among CD3^+^ T cells in the indicated organs is shown (*n* = 4 per strain). Bar graph depicts means ± SD. Vα19: Vα19 mice, Vβ8: Vβ8 mice, B6: C57BL/6. * *p* < 0.05, ** *p* < 0.01, *** *p* < 0.005, **** *p* < 0.0001, ns: not significant (One-way ANOVA).

**Figure 3 biomedicines-12-00137-f003:**
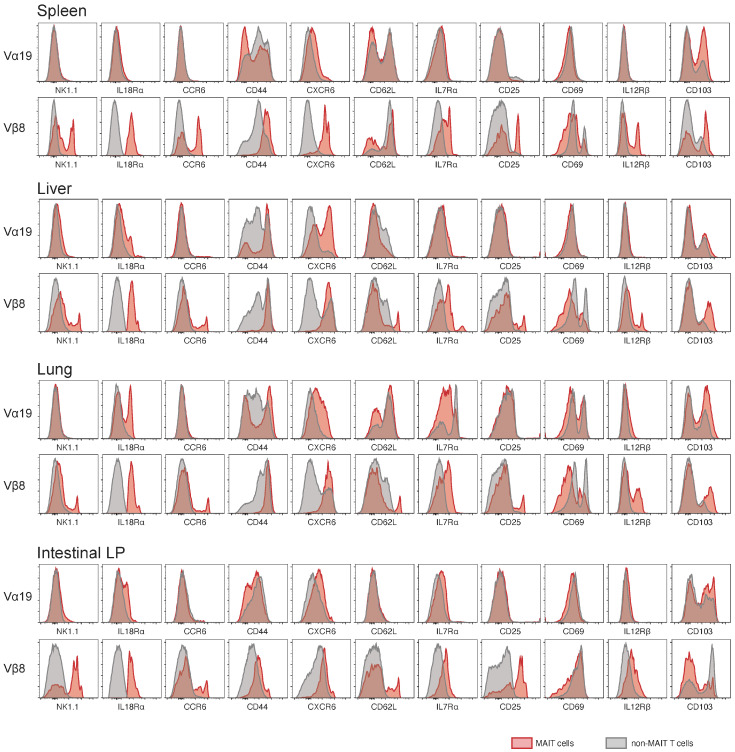
Comparative phenotypic analysis of MAIT cells and non-MAIT T cells. Expression levels of the indicated markers in MAIT cells (shaded in red) and in non-MAIT T cells (shaded in grey) from the spleen, liver, lung, and intestine of Vα19 and Vβ8 mice are depicted as a histogram. Data are representative of three independent experiments (*n* = 3 per strain).

**Figure 4 biomedicines-12-00137-f004:**
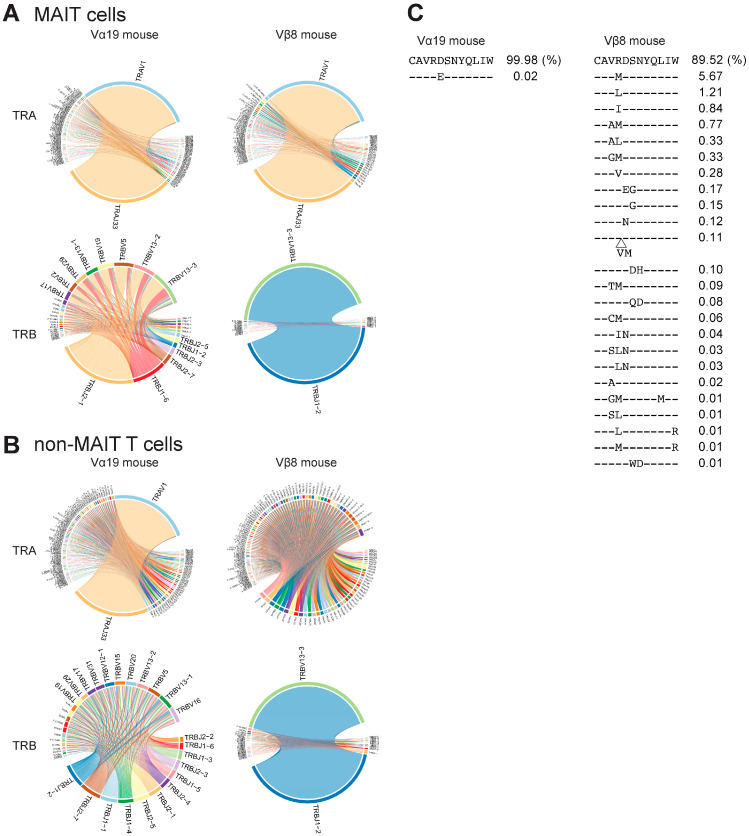
TCR repertoire analysis. (**A**) Circos plot for MAIT cells. Circos plots for MAIT cells in the thymocytes from the indicated mouse strain are depicted. Upper and lower panels indicate TRA and TRB, respectively. (**B**) Circos plot for non-MAIT T cells. Circos plots for non-MAIT T cells in the thymocytes from the indicated mouse strain are shown. Upper and lower panels indicate TRA and TRB, respectively. (**C**) Diversity in CDR3α. Amino acid sequences of CDR3α are aligned for MAIT cells in Vα19 mice (**left panel**) and those in Vβ8 mice (**right panel**). -; the same amino acid indicated in the first line of sequences. Δ; additional sequence insertion point. The percentage shows the frequency of the clone(s) harboring the indicated CDR3 sequence among MAIT cells.

**Figure 5 biomedicines-12-00137-f005:**
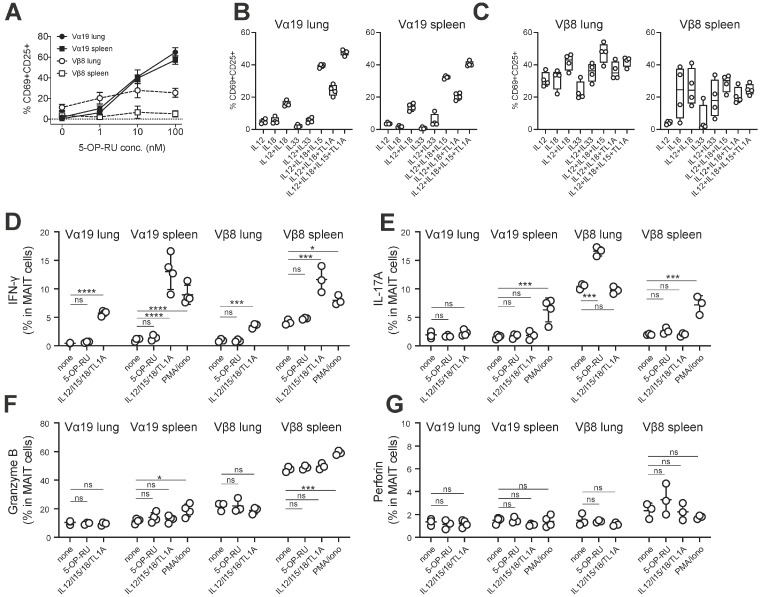
MAIT cell activation and production of IFN-γ and IL17A. (**A**) Activation of MAIT cells by 5-OP-RU. Percentages of CD69^+^CD25^+^ cells in MAIT cells among lung mononuclear cells and the spleen cells from the indicated mice followed by stimulation with varying amounts of 5-OP-RU (1–100 nM) are depicted (*n* = 4 per group). (**B**,**C**) Activation of MAIT cells by cytokines. Percentages of CD69^+^CD25^+^ cells in MAIT cells among lung mononuclear cells and the spleen cells from Vα19 mice (**Β**) and Vβ8 mice (**C**) followed by stimulation with the combination of the indicated cytokine(s) are depicted (*n* = 4 per group). (**D**–**G**) Intracellular cytokine staining. Frequency of MAIT cells expressing IFN-γ (**D**), IL-17A (**E**), granzyme B (**F**), and perforin (**G**) upon the indicated stimuli is shown (*n* = 4 for Vα19 mouse, *n* = 3 for Vβ8 mouse). * *p* < 0.05, *** *p* < 0.001, and **** *p* < 0.0001, ns: not significant. Vα19: Vα19 mouse, Vβ8: Vβ8 mouse.

**Figure 6 biomedicines-12-00137-f006:**
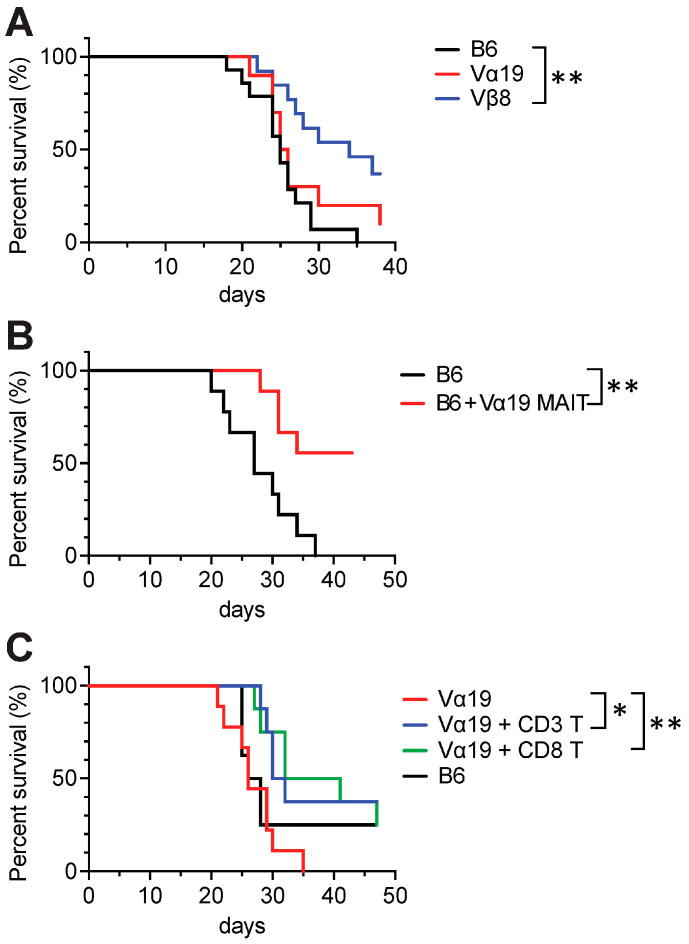
Tumor resistance of Vα19 and Vβ8 mice. (**A**) Superiority of Vβ8 mice in conferring survival extension. Survival of the indicated strains of the mice was monitored upon tumor inoculation (3 × 10^5^ LLC, I. V.); ** *p* < 0.01 (*n* = 14 per strain, log-rank test). (**B**) Survival superiority conferred by Vα19 MAIT cells. Survival of the control mice (B6) and of the control mice that received MAIT cells prepared from Vα19 mice (B6 + Vα19 MAIT) were monitored as per (**A**). ** *p* < 0.01 (*n* = 10 per group, log-rank test). (**C**) Survival superiority conferred to Vα19 mice by CD8^+^ T cells. Survival of the control mice (B6), Vα19 mice (Vα19), and Vα19 mice that received either CD3^+^ T cells (Vα19 + CD3 T, 1.0 × 10^6^ cells) or CD8^+^ T cells (Vα19 + CD8 T 1.0 × 10^6^ cells) from the control mice was monitored as per (**A**). * *p* < 0.05, ** *p* < 0.01 (*n* = 10 per group, log-rank test).

**Table 1 biomedicines-12-00137-t001:** Characterization of iPSCs and chimeric mice used for germline transmission.

iPS Clones	TCRα		TCRβ			Νο. of Chimeric Male (Animal ID) (% of Donor Coat Color)		
	Vα	Jα	Vβ	D	Jβ	(60–90%)	(30–60%)	(<30%)
L7	*Trav1* (Vα19)	*Traj33* (Jα33)	*Trbv13-3*01* (Vβ8)	*Trbd1*01*	*Trbj1-2*01*	2 (#25, 26)	2 (#27, 28)	1 (#13)
L11	*Trav1* (Vα19)	*Traj3*3 (Jα33)	*Trbv19*01*, **03* (Vβ6)	*Trbd2*01*	*Trbj2-3*01*	2 (#29, 30)	2 (#31, 32)	-
L19	*Trav1* (Vα19)	*Traj33* (Jα33)	*Trbv19*01*, **03* (Vβ6)	*Trbd2*01*	*Trbj2-3*01*	-	-	2 (#33, 34)

The iPSC clones used for chimeric mouse generation are indicated in the left column with the corresponding usage of Vα, Jα, Vβ, Dβ, and Jβ. The number of chimeric mice harboring 60–90% of the chimerism is shown with the proper identifier (#) in the right column.

## Data Availability

The original contributions presented in the study are included in the article/Appendix A, further inquiries can be directed to the corresponding author. The RNA-seq data for the MAIT TCR repertoire analysis have been deposited with links to BioProject accession number PRJDB15105 in the DDBJ BioProject database.

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
