# Peer review of "Mice Generated with Induced Pluripotent Stem Cells Derived from Mucosal-Associated Invariant T Cells"

_biomedicines, 2024, doi:10.3390/biomedicines12010137_

Round 1

Reviewer 1 Report

Comments and Suggestions for Authors

The manuscript by Sugimoto et al. reports new mouse strains that were produced via induced pluripotent stem cells derived from MAIT cells. In doing so, the iPS cells have pre-rearranged T cell receptor genes. MAIT cells are numerous in humans but rare in most laboratory mouse strains. Previous attempts in increasing the frequency of MAIT cells in mice using TCR transgenics led to experimental artifacts due to poor control of TCR rearrangement timing during T cell development. Thus, these new mouse models are of great interest.

The authors generated 3 different lines, either expressing a TCRb chain derived from a MAIT cell, a TCRa chain derived from a MAIT cell, or both. The analysis of the first 2 lines is presented in this manuscript. Expression of the TCRb chain increased the frequency of MAIT cells in the spleen from 0.1% to 0.5%, while expression of the TCRa chain increased this frequency to 36% of the T cells. The MAIT cells in these animals have the expected phenotype of MAIT cells, which provide a new and interesting model to be able to study this population. Furthermore, the authors show that mice with the TCRb chain have an increased survival time upon inoculation with lung carcinoma cell line, while no difference is observed in mice with the TCRa chain. Furthermore, the authors showed that other T cell subsets of immune cells are required for MAIT cells to exert anti-tumor activity.

The experiments are properly executed, presented and controlled. The results are certainly interesting and deserve publication, although they also raise some new and important questions that are not addressed.

Comments:   Due to the primary role of in the TRAV1-TRAJ33 chain of the MAIT TCR in interacting with the antigen/MR1 complex, it would be expected that mice having pre-rearranged TRAV1-TRAJ33 or TRAV1-TRAJ33/TRBV13-3-TRDJ1-2 would have increased frequency of MAIT cells. However, surprisingly, T cells that are not MAIT cells are also developing in these mice. How is this happening? Are the non-MAIT cells actually expressing TRAV1-TRAJ33 but paired with TCRb chains that are non-permissive of MR1 recognition? This would potentially suggest that TRAV1-TRAJ33 is not restricted to MR1 recognition and that T cells bearing this TCRa chain can be positively selected by other MHC molecules. Crossing of TRAV1-TRAJ33 with b2m-deficient mice would be informative in that matter. Analysis if CDR3 regions for TCRb chains commonly used by MAIT and non-MAIT in these mice would also be useful. Are these cells still expressing TRAV1-TRAJ33 and selected by MR1 but not stained with the 5-OP-RU/MR1 tetramer? This could be tested by crossing the TRAV1-TRAJ33 mice to MR1-deficient mice. Are these cells using a different TCR alpha chain? In such instance, what would it mean in terms of allelic exclusion? Although the authors decided to not characterize further the TRAV1-TRAJ33/TRBV13-3-TRDJ1-2 mice, the same questions are even more pressing in such setting where both chains are now pre-rearranged.

The authors show that the frequency (what is the legend of the y axis on figure 2 D and F?) of gd T cells is unaffected in mice with increased MAIT frequency. The actual cell number for each population should be calculated and displayed. These results somehow contradict recent data showing an increase in MAIT cells in mice deficient for gd T cells or gd T cells and NKT cells. This should be discussed.

Minor comments: on line 160, Vb8 mice expressed higher levels of what?

Comments on the Quality of English Language

n/a

Author Response

The manuscript by Sugimoto et al. reports new mouse strains that were produced via induced pluripotent stem cells derived from MAIT cells. In doing so, the iPS cells have pre-rearranged T cell receptor genes. MAIT cells are numerous in humans but rare in most laboratory mouse strains. Previous attempts in increasing the frequency of MAIT cells in mice using TCR transgenics led to experimental artifacts due to poor control of TCR rearrangement timing during T cell development. Thus, these new mouse models are of great interest.

The authors generated 3 different lines, either expressing a TCRb chain derived from a MAIT cell, a TCRa chain derived from a MAIT cell, or both. The analysis of the first 2 lines is presented in this manuscript. Expression of the TCRb chain increased the frequency of MAIT cells in the spleen from 0.1% to 0.5%, while expression of the TCRa chain increased this frequency to 36% of the T cells. The MAIT cells in these animals have the expected phenotype of MAIT cells, which provide a new and interesting model to be able to study this population. Furthermore, the authors show that mice with the TCRb chain have an increased survival time upon inoculation with lung carcinoma cell line, while no difference is observed in mice with the TCRa chain. Furthermore, the authors showed that other T cell subsets of immune cells are required for MAIT cells to exert anti-tumor activity.

The experiments are properly executed, presented and controlled. The results are certainly interesting and deserve publication, although they also raise some new and important questions that are not addressed.

Comments:   Due to the primary role of in the TRAV1-TRAJ33 chain of the MAIT TCR in interacting with the antigen/MR1 complex, it would be expected that mice having pre-rearranged TRAV1-TRAJ33 or TRAV1-TRAJ33/TRBV13-3-TRDJ1-2 would have increased frequency of MAIT cells.

Answer:

Thank you for your comments. We interpret our data that rearranged TRAV1-TRAJ33, TRBV13-3-TRDJ1-2, and TRAV1-TRAJ33/TRBV13-3-TRDJ1-2 had an impact on T cell development in the thymus. However, we are not sure whether other rearranged TRAV and/or TRBV would give the same results (increased frequency of T cells harboring rearranged TRAV and/or TRBV in question). We speculate that there exist some unknown mechanism(s) that impede further rearrangement in the case for TRAV1-TRAJ33. Otherwise, it is expected that rearranged TRAV1-TRAJ33 would easily be substituted by other combination of TRAV and TRAJ, resulting in little change in the frequency of MAIT cells.

However, surprisingly, T cells that are not MAIT cells are also developing in these mice. How is this happening?

Answer:

Thank you for this question. In brief, the answer is that T cells (not MAIT cells) are developed from the other allele. The explanation is found in the text as follows” As Trav1 is located at the most distal region of the TCRα locus, once MAIT cell-specific TCRα (Trav1-Traj33) was selected, iNKT cell-specific TCR (Trav11-Traj18) and/or any other Trav-Traj rearrangement could not occur in the same allele. Thus, it is plausible that V-J recombination from the other allele is at work in these mice to ensure TCRα chain diversity to a certain degree, thus allowing iNKT cell and γδ T cell development.”

(Discussion p13, lines 21-p14 lines 2).

Are the non-MAIT cells actually expressing TRAV1-TRAJ33 but paired with TCRb chains that are non-permissive of MR1 recognition?

Answer:

Our TCR repertoire analysis could only detect the transcripts in bulk and is not a single cell-based one. Thus, we cannot answer that question. However, should this happen, one would not call these cells as MAIT cells.

This would potentially suggest that TRAV1-TRAJ33 is not restricted to MR1 recognition and that T cells bearing this TCRa chain can be positively selected by other MHC molecules.

Answer:

So far as we understand, MAIT cells are defined as T cells harboring TRAV1-TRAJ33 whose development is contingent upon MR1. However, we cannot formally exclude the possibility that T cells harboring TRAV1-TRAJ33 whose development is dependent on MHC molecules other than MR1 would exist.

Crossing of TRAV1-TRAJ33 with b2m-deficient mice would be informative in that matter.

Answer:

Thank you for your suggestion. We would consider to perform the experiment in the future.

Analysis if CDR3 regions for TCRb chains commonly used by MAIT and non-MAIT in these mice would also be useful.

Answer:

Thank you for your suggestion. As far as we have searched, we could not find common CDR3b in MAIT cells and non-MAIT T cells (see Table S1, TRBV-Va19 MAIT vs TRBV-Va19 non-MAIT).

Are these cells still expressing TRAV1-TRAJ33 and selected by MR1 but not stained with the 5-OP-RU/MR1 tetramer?

This could be tested by crossing the TRAV1-TRAJ33 mice to MR1-deficient mice.

Answer:

MAIT cells are defined as T cells harboring TRAV1-TRAJ33 whose development is contingent upon MR1 which can be detected with 5-OP-RU/MR1 tetramer. Thus, if such cells exist, we would not call them MAIT cells.

We also thank you for your suggestion for the mice. We would consider to perform the experiment in the future.

Are these cells using a different TCR alpha chain? In such instance, what would it mean in terms of allelic exclusion?

Answer:

If you ask on 5-OP-RU/MR1 tetramer negative cells (non-MAIT T cells), the answer is yes. The TRAV repertoire analysis on non-5-OP-RU/MR1 tetramer+ cells (considered to be non-MAIT T cells) has revealed many different combinations of TRAV-TRAJ (compare TRAs in Figure 4A and 4B, and Tables S1). While these data are transcripts for 5-OP-RU/MR1 tetramer+ cells (MAIT cells) and those for 5-OP-RU/MR1 tetramer negative cells (non-MAIT T cells), one can see an increase in the diversity of Trav-Traj (mRNA) in non-MAIT T cells (Fig. 4B TRA) relative to MAIT cells (Fig. 4A TRA). 

It is difficult to comment on allelic exclusion on TRAV, though a net increase in Trav1-Traj33 (transcript) has been observed in 5-OP-RU/MR1 tetramer+ cells (MAIT cells) relative to 5-OP-RU/MR1 tetramer negative cells (non-MAIT T cells). While allelic exclusion ensures the only one productively rearranged allele is expressed on the surface of B cell or T cell, our present TCR repertoire analysis has focused on the transcripts in bulk but not on the TCR protein on the cell surface. Thus, it is difficult to give sound answer to the question, as we have no data on TCR repertoire in protein at a single cell level.

Although the authors decided to not characterize further the TRAV1-TRAJ33/TRBV13-3-TRDJ1-2 mice, the same questions are even more pressing in such setting where both chains are now pre-rearranged.

 Answer:

Thank you for your suggestion. We would analyze Vα19/Vβ8 mice in the next work.

The authors show that the frequency (what is the legend of the y axis on figure 2 D and F?) of gd T cells is unaffected in mice with increased MAIT frequency. The actual cell number for each population should be calculated and displayed.

 Answer:

Thank you for your suggestion. We basically agree with you. However, we have only measured the frequency of each subset relative to the total CD3+ cells (considered to be 100%), but not the cell number in this study. We will consider to perform such enumeration in the future.

These results somehow contradict recent data showing an increase in MAIT cells in mice deficient for gd T cells or gd T cells and NKT cells. This should be discussed.

Answer:

Thank you for your comment. We respectfully disagree with the reviewer’s opinion that “these results somehow contradict recent data”. However, we find it important to add such a reference for discussion. When mice devoid of some subsets of T cells (innate-like T cells) are generated, it is plausible that a part of “niche” for these cells becomes available for another subset (in these cases, MAIT cells) to develop favorably, which in turn results in an increase of MAIT cells. Accordingly, we have added the following sentences in the Discussion section. “It is noteworthy that mice lacking γδ T cells and/or iNKT cells show an increase in MAIT cells (Xu et al., Mucosal Immunol 2023 Aug;16(4):446-461. doi: 10.1016/j.mucimm.2023.05.003.). However, under the normal circumstances, it is likely that there is competition between MAIT cells and these cells for the niche. Thus, the loss of γδ T cells and/or iNKT cells would provide a niche for MAIT cells to develop favorably.

Minor comments: on line 160, Vb8 mice expressed higher levels of what?

Answer:

We are sorry for the inconvenience. To be clearer, we revised the text as follows: MAIT cells in Vβ8 mice expressed higher levels of the surface molecules except for CD62L, (page 9, line4)

Reviewer 2 Report

Comments and Suggestions for Authors

the work is potentially interesting

however, all 3 pictures that are described are not visible in the work at all

forward work with pictures

Only supplementary figures with monoclonal antibodies that were used are shown in the paper

1. changes in the introduction: it is necessary to point out that the cells of innate immunity are divided into several groups based on the immunophenotype as well as on the basis of the produced cytokines, as shown in the previous works that should be mentioned: PMID: 31079327 , PMID: 37296617

2. add in the discussion that cytokines such as IL-2, IL-12, IL-15 also have effects on NK cells and other cytotoxic lymphocytes that were examined here and compare the effects with findings from other studies in humans with those obtained on I think model: PMID: 29948616

Comments on the Quality of English Language

need corections

Author Response

the work is potentially interesting

however, all 3 pictures that are described are not visible in the work at all

forward work with pictures

Only supplementary figures with monoclonal antibodies that were used are shown in the paper

Answer: Thank you for appreciating our work. However, it was not clear for us what the reviewer means “all 3 pictures and figures with monoclonal antibodies”. To avoid any problem, we will check all the figures before upload.

  1. changes in the introduction: it is necessary to point out that the cells of innate immunity are divided into several groups based on the immunophenotype as well as on the basis of the produced cytokines, as shown in the previous works that should be mentioned: PMID: 31079327, PMID: 37296617

Answer:

Thank you for your suggestion. We added PMID: 31079327 (discusses the different member of innate lymphocytes (ILC1, ILC2, and ILC3)) in the Introduction. We found that PMID: 37296617 focuses on NK cell development, which is not relevant to the present work. Instead, we added another reference in the same sentence (Mazzurana et al., Seminars in Immunopathology (2018) 40:407–419).

  1. add in the discussion that cytokines such as IL-2, IL-12, IL-15 also have effects on NK cells and other cytotoxic lymphocytes that were examined here and compare the effects with findings from other studies in humans with those obtained on I think model: PMID: 29948616

Answer:

Thank you for your suggestion. While we have a difficulty to understand what “compare the effects with findings from other studies in humans with those obtained on I think model: PMID: 29948616” means, we have added the reference and modified the text in the discussion as follows “In this respect, it will be interesting to examine whether IL-2 and IL-15 have an effect on NKG2D and CD158a-b in MAIT cells, because the cytokines enhance NKG2D expression in different lymphocytes and appear to increase anti-tumor potential of NK cells (Velutic et al., Pathology & Oncology Research (2020) 26:223–231).

Remark:

Although the reviewer pointed out that English should extensively be edited, the text had already been checked and edited by a professional English Editor (please see the attached certificate).

Reviewer 3 Report

Comments and Suggestions for Authors

In the manuscript entitled "Mice generated with induced pluripotent stem cells derived from mucosal-associated invariant T cells", Sugimoto et al present many observations of genetically modified mice they generated which bear recombined TCRa and TCRb alleles normally found in mucosal-associated invariant T (MAIT) cells. The work is very interesting and of obvious scientific value, and although it opens many more questions than it answers, this work serves as an important first report on a new mouse strain worthy of future studies. The main downside of the manuscript is that it is written so densely, that a reader without a strong background on MAIT cell biology is unlikely to find this report of much interest or use. Authors could improve the manuscript by providing more introduction and context throughout the manuscript. 

Major criticisms:

(1) Through embryo microinjection of MAIT-iPSCs, authors generated chimeric mice that were then backcrossed to create 3 lines of Va19, Vb8, and Va19/Vb8 mice. This process is a bit confusing, a visual diagram of the chimeric crossings, backcrossing, and outcrossing would be helpful to clarify this process for the reader. 

(2) In figure 1, authors give some preliminary data from mice harboring both Va19 and Vb8 TCR but no more data is provided regarding this strain in the rest of the manuscript. It would seem that more analyses of this completely recombinant TCR transgenic animal would provide valuable insights into the effects of TCR expression on the differentiation of various T cell subsets. Comparison of this strain with other existing MAIT TCR-transgenic animals would be quite interesting as well. The rationale for not including these mice (even if it is technical or practical in nature) should be clarified in the text. 

(3) Authors present interesting data on the frequency, marker expression profile, and TCR gene rearrangements of various T cell subsets in these animals in Figure 2-4. This data is very interesting but is not adequately discussed in the manuscript. For example, the Va19 mice show many more MAIT cells than Vb8 or WT mice and data is provided in regards to TCR gene recombinations in MAIT and non-MAIT cells of these mice in Figure 4, but no mechanism is proposed for how a divergence in MAIT cell numbers arises. An attempt to unify cellular and genetic observations would be very helpful to improving the clarity of the manuscript. 

(4) Functional data of MAIT cells derived from Va19 or Vb8 mice is provided in Figure 5. Again, this seems to be quite an important observation, but it is lost within the other data of the paper. From this data, it would appear that Vb8 MAIT cells are mostly non-functional with respect to TCR-dependent responses. The diminished functional response of Vb8 MAIT cells to 5-OP-RU is described as a "plateau" in the text, but would appear not to really change from unstimulated cells at all. This perhaps would be best described as non-functional with regards to TCR-dependent responses. 

This result seems particularly surprising given that Va19 MAIT cells show a diverse TCR repertoire while Vb8 MAIT cells are almost completely bearing a canonical MAIT-TCR (as per Figure 4).  Given this genetic data, one could predict diminished functionality within Va19 relative to Vb8. Authors should identify and attempt to resolve this contradiction in the discussion section. 

(5) In the final figure of the manuscript, authors present data on the ability of Va19, Vb8, or WT mice to resist tumor challenge. This data is again quite interesting, but given the extent of immune cell changes in the mice is difficult to draw strong conclusions from. The experiments are sensibly designed, and in particular the transfer of Va19 MAIT cells to WT mice indicates that this may open the doors to interesting models of cancer cell therapy. That being said, this data does not necessarily add greatly to this specific manuscript and authors might consider moving this data to supplemental. A more targeted follow up study using adoptive transfer of Va19 MAIT cells as a murine model of T-cell cancer immunotherapy could be highly interesting. 

Overall the study is highly interesting and should be published but authors are strongly encouraged to edit the manuscript for clarity and to address the criticisms raised above. 

Other minor/specific critiques:

(1) Line 100 - Inclusion of pictures of the chimeric mice with varying levels of chimerism as supplemental data might be helpful to contextualize the data provided in Table 1. 

(2) Figure 3 - An in figure legend would be helpful to identify the MAIT (red) and non-MAIT (grey) histograms. 

(3) Figure 4 - As above, an in figure label indicating that 4A shows MAIT cell TCRs whereas 4B shows non-MAIT cell TCRs

(4) Figure 5A - Inclusion of control cells from a WT mouse would be helpful if such an experiment is possible? If not, perhaps including iPSC derived MAIT cells would work as well. Including such a control would help a reader better understand the functional observations presented here. 

(5) Line 330 - The line "novel mice derived from iPSCs generated from lung cells are reported" could be interpreted to state that such novel mice have been previously reported. Authors should make clear that these mice are "reported here"

(6) Line 369 - Authors reference the data here as evidence that MAIT cells require CD8+T cells for anti-metastatic function. While this is supported by the data, it does not necessarily preclude a role for NK cells. Authors should rewrite this line to make it clear that both NK and CD8+ T cell interaction with the MAIT cells may be implicated. 

Author Response

In the manuscript entitled "Mice generated with induced pluripotent stem cells derived from mucosal-associated invariant T cells", Sugimoto et al present many observations of genetically modified mice they generated which bear recombined TCRa and TCRb alleles normally found in mucosal-associated invariant T (MAIT) cells. The work is very interesting and of obvious scientific value, and although it opens many more questions than it answers, this work serves as an important first report on a new mouse strain worthy of future studies. The main downside of the manuscript is that it is written so densely, that a reader without a strong background on MAIT cell biology is unlikely to find this report of much interest or use. Authors could improve the manuscript by providing more introduction and context throughout the manuscript. 

Answers: Thank you for the comments and critiques. As suggested by the reviewer, we have added following text in Introduction to help readers appreciate the present paper. “Mouse studies are an effective approach to elucidating the function of MAIT cells in health and disease. Nonetheless, mouse MAIT cells are rare, representing 1/10~1/1000 of the human cells, making it difficult to study their functions (Godfrey et al., 2015; Godfrey et al., 2019; Toubal et al., 2019)"}" id="-1823498801">(Godfrey et al., 2015). “ (Introduction page 4, lines 8-11)

Major criticisms:

  • Through embryo microinjection of MAIT-iPSCs, authors generated chimeric mice that were then backcrossed to create 3 lines of Va19, Vb8, and Va19/Vb8 mice. This process is a bit confusing, a visual diagram of the chimeric crossings, backcrossing, and outcrossing would be helpful to clarify this process for the reader. 

Answers: Thank you for the critiques. Accordingly, we now provide a revised figure to clarify the procedure to create Vα19, Vβ8, and Vα19/Vβ8 mice for clarity (please see revised Figure S1A-B, and E), as suggested by the reviewer.

  • In figure 1, authors give some preliminary data from mice harboring both Va19 and Vb8 TCR but no more data is provided regarding this strain in the rest of the manuscript. It would seem that more analyses of this completely recombinant TCR transgenic animal would provide valuable insights into the effects of TCR expression on the differentiation of various T cell subsets. Comparison of this strain with other existing MAIT TCR-transgenic animals would be quite interesting as well. The rationale for not including these mice (even if it is technical or practical in nature) should be clarified in the text. 

Answers: Thank you for the comments and critiques. Basically, we agree with the reviewer that use of Vα19/Vβ8 mice throughout the study is important for shedding much light on the function of MAIT cells. The incomplete nature of the present study is mainly due to the fact that the study focused on the effect of rearranged configuration of either TRAV or TRBV specific for MAIT cells. Given that the presence of in-frame rearranged TRBV in the allele results in allelic exclusion as shown in Figure 4, we sought whether such TRBV impacted the generation of MAIT cells in vivo. Similarly, we sought whether in-frame rearranged TRAV (so-called invariant) in the allele also affected MAIT cell development. However, we avoided examining MAIT cell function in the presence of both in-frame rearranged TRAV and TRBV because the presence of TRBV would limit the repertoire of TRBV to only TRAV13-3 and thus the repertoire and function of MAIT cells. We agree with the reviewer that comparative studies with MAIT-TCR transgenic and Vα19/Vβ8 mice will help elucidating the function of MAIT cells further as the reviewer suggested. Accordingly, we have added the following sentences in Results, “Hereafter, all experiments were performed with Vα19 and Vβ8 mice, since the presence of both in-frame rearranged Trav1-Traj33 and Trbv13-d1-j1-2 as seen in Vα19/Vβ8 mice may limit the repertoire and function of MAIT cells.” (Results Page 7, lines 5-6)

  • Authors present interesting data on the frequency, marker expression profile, and TCR gene rearrangements of various T cell subsets in these animals in Figure 2-4. This data is very interesting but is not adequately discussed in the manuscript. For example, the Va19 mice show many more MAIT cells than Vb8 or WT mice and data is provided in regards to TCR gene recombinations in MAIT and non-MAIT cells of these mice in Figure 4, but no mechanism is proposed for how a divergence in MAIT cell numbers arises. An attempt to unify cellular and genetic observations would be very helpful to improving the clarity of the manuscript. 

Answer: Thank you for your comments. Based on the reviewer suggestion, we have added the following sentences in Discussion. “The presence of rearranged Trav1-Traj33 in all cell alleles, including immature T cells such as MAIT cell precursors (which possess the Trav1-Traj33 transcript but have not yet been selected by MR1), favors biased selection of MAIT cells because there are many cells expressing the transcript during T cell development in Vα19 mice. Given that the selection of Trav and Traj DNA fragments is random during TCRα gene rearrangement, and that only a small population of precursor cells harboring Trav1-Traj33 is selected by MR1 in WT mice, this abundance of cells harboring the Trav1-Traj33transcript would result in far more MAIT cells in Vα19 mice than in WT mice. In contrast, the number of MAIT cells is relatively low in Vβ8 mice. This is in part due to the absence of the rearranged Trav1-Traj33 in the allele of Vβ8 mouse. However, the presence of a rearranged Trbv13-3-trbj1-2 in the allele derived from the MAIT cell TCR would allow preferential and/or privileged pairing with nascent MAIT cells harboring Trav1-Traj13. Such pairing is further enhanced by allelic exclusion imposed by Trbv13-3-trbj1-2. These features results in an increase in MAIT cells in Vβ8 mice compared to WT mice (Discussion page 14, lines 8-22).

(4) Functional data of MAIT cells derived from Va19 or Vb8 mice is provided in Figure 5. Again, this seems to be quite an important observation, but it is lost within the other data of the paper. From this data, it would appear that Vb8 MAIT cells are mostly non-functional with respect to TCR-dependent responses. The diminished functional response of Vb8 MAIT cells to 5-OP-RU is described as a "plateau" in the text, but would appear not to really change from unstimulated cells at all. This perhaps would be best described as non-functional with regards to TCR-dependent responses. 

This result seems particularly surprising given that Va19 MAIT cells show a diverse TCR repertoire while Vb8 MAIT cells are almost completely bearing a canonical MAIT-TCR (as per Figure 4).  Given this genetic data, one could predict diminished functionality within Va19 relative to Vb8. Authors should identify and attempt to resolve this contradiction in the discussion section. 

Answer: We appreciate the reviewer critiques regarding the possible dysfunction of MAIT cells in the mice. However, we respectfully disagree with the reviewer. From the data shown in Figure 5A and Figure S3, MAIT cells in Vβ8 mice are already activated in the absence of any external stimuli, most likely reflecting MAIT cell phenotype as effector cells being stimulated in vivo. This is also supported by expression of Granzyme B in the lung and the spleen MAIT cells from Vβ8 mice in the absence of any stimuli (Figure 5F). These features reflect the phenotype of MAIT cells as effector cells in vivo. It is thus plausible that seeming failure of 5-OP-RU to activate MAIT cells ex vivo could be due to MAIT cell phenotype as effector cells in vivo.

To avoid any confusion, however, we have added the following sentences in Discussion. “The fact that MAIT cells in Vβ8 mice showed activated phenotype and produced Granzyme B, an effector molecule, in the absence of external stimuli ex vivo may indicate that MAIT cells are present as effector cells in vivo (Fig. 5A, 5F and S3). This may in part explain why 5-OP-RU only weakly activated Vβ8 MAIT cells ex vivo (Fig. 5A).” (Discussion, Page 16, lines 6-10).

Regarding the diminished functionality of MAIT cells in Vα19 mice relative to Vβ8 mice, we are a little concerned because the reviewer seems to have misunderstood the term somewhat. In this paper, we refer MAIT cells to be mMR1-tet+TCRβ+ cells. We showed that mMR1-tet+TCRβ+ cells in Vα19 mice comprise mainly the well-known combination of TRAV1-TRAJ33 as seen in Figure 4A. However, it is noteworthy that among all V-J combinations found in mMR1-tet+TCRβ+ cells (considered to be MAIT cells), the frequency of TRAV1-TRAJ33 and the diversity of TRAV-TRAJ combinations other than TRAV1-TRAJ33 are almost identical between Vα19 mice and Vβ8 mice (Figure 4A, TRA). Thus, it is hard to conclude that Vα19 MAIT cells (MAIT cells in Vα19 mice) show a diverse TCR repertoire and exhibit diminished functionality compared to those in Vβ8 mice.  

  • In the final figure of the manuscript, authors present data on the ability of Va19, Vb8, or WT mice to resist tumor challenge. This data is again quite interesting, but given the extent of immune cell changes in the mice is difficult to draw strong conclusions from. The experiments are sensibly designed, and in particular the transfer of Va19 MAIT cells to WT mice indicates that this may open the doors to interesting models of cancer cell therapy. That being said, this data does not necessarily add greatly to this specific manuscript and authors might consider moving this data to supplemental. A more targeted follow up study using adoptive transfer of Va19 MAIT cells as a murine model of T-cell cancer immunotherapy could be highly interesting. 

Answer: Thank you for your appreciation for our data. While the reviewer suggests to put the data to supplemental, we consider it to an integral part of the manuscript describing some nature of Vα19 mice and Vβ8 mice. In particular, as the reviewer pointed out, adoptive transfer experiments of Vα19 MAIT cells into WT mice open up novel avenues for cancer immunotherapy. Thus, we strongly believe that data showing the tumor resistance of Vβ8 mice and Vα19 mice supplemented with CD3 or CD8 T cells from WT mice should be kept here.

Overall the study is highly interesting and should be published but authors are strongly encouraged to edit the manuscript for clarity and to address the criticisms raised above. 

Other minor/specific critiques:

  • Line 100 - Inclusion of pictures of the chimeric mice with varying levels of chimerism as supplemental data might be helpful to contextualize the data provided in Table 1. 

Answer: Basically, we agree with the reviewer. Unfortunately, we do not have such pictures. 

  • Figure 3 - An in figure legend would be helpful to identify the MAIT (red) and non-MAIT (grey) histograms. 

Answer: As suggested, we have identified MAIT cells and non-MAIT T cells in Figure 3.

  • Figure 4 - As above, an in figure label indicating that 4A shows MAIT cell TCRs whereas 4B shows non-MAIT cell TCRs

Answer: As suggested, we have indicated MAIT cells and non-MAIT T cells in Figure 4A and 4B, respectively.

  • Figure 5A - Inclusion of control cells from a WT mouse would be helpful if such an experiment is possible? If not, perhaps including iPSC derived MAIT cells would work as well. Including such a control would help a reader better understand the functional observations presented here. 

Answer: Thank you for your fruitful comments. We agree with the reviewer and would envision to perform such experiments in the near future.

  • Line 330 - The line "novel mice derived from iPSCs generated from lung cells are reported" could be interpreted to state that such novel mice have been previously reported. Authors should make clear that these mice are "reported here"

Answer: As suggested, we have revised the text as follows; In addition to these mice, novel mice derived from iPSCs generated from lung MAIT cells are reported here (Discussion page 14, line 6).

  • Line 369 - Authors reference the data here as evidence that MAIT cells require CD8+T cells for anti-metastatic function. While this is supported by the data, it does not necessarily preclude a role for NK cells. Authors should rewrite this line to make it clear that both NK and CD8+ T cell interaction with the MAIT cells may be implicated. 

Answer: We agree with the reviewer. As suggested, we have revised the text as follows; This indicates that both NK and CD8+T cell interaction with MAIT cells may be implicated in anti-metastatic activity (Discussion page 16, lines 17-19).

Round 2

Reviewer 2 Report

Comments and Suggestions for Authors In the material and methods section:
1. add in the experimental animals section: line 435 how many mice were used in each experiment, because it does not say only the age of the mice, as in the case of chimeric mice (line 444), how many experimental mice were used in each tested group as well as per experiment and that there were repetitions of the experiment.
2. At the end of the section flow cytometry for the examination of cells as well as for the examination of intracellular cytokines, add a reference for the method as previously suggested and described in the paper: PMID: 33131355
3. In the flow cytometry section, it is not necessary to describe the number of lasers for the device, it is known for a specific model. However, it should be added in that section that the combinations of antibodies are shown in the table and the table should be placed in the supplementary data and not in the main findings of the paper, because the paper already has a lot of data and pictures, and it is not a finding but the chemicals that were used, which is usually listed in the text without additional tables and describe in words.
4. For the data analysis method shown in Figure 4, add in the statistical data processing section either a reference or indicate which program was used (Visualisation with Circos) because it is not a program for classical statistical data processing. It is not mentioned anywhere in the material and method section. It is necessary to add the analysis to the material and method section.       Comments on the Quality of English Language

ok

Author Response

Answers to reviewer 2 (2nd round)

In the material and methods section:
1. add in the experimental animals section: line 435 how many mice were used in each experiment, because it does not say only the age of the mice, as in the case of chimeric mice (line 444), how many experimental mice were used in each tested group as well as per experiment and that there were repetitions of the experiment.

Answer: The number of the mice used in each experiment is already shown in the corresponding figure legends. However, to be clearer, we have added to “n=3 per strain” in the figure legend 1 and 3.

  1. At the end of the section flow cytometry for the examination of cells as well as for the examination of intracellular cytokines, add a reference for the method as previously suggested and described in the paper: PMID: 33131355

Answer: Given the generality of the method for flow cytometry and intracellular cytokine staining, we do not agree with reviewer in that such a particular paper should be cited here.

  1. In the flow cytometry section, it is not necessary to describe the number of lasers for the device, it is known for a specific model. However, it should be added in that section that the combinations of antibodies are shown in the table and the table should be placed in the supplementary data and not in the main findings of the paper, because the paper already has a lot of data and pictures, and it is not a finding but the chemicals that were used, which is usually listed in the text without additional tables and describe in words.

Answer: We respectfully disagree with reviewer that antibodies used in the experiments should be listed in the supplemental table. We have already listed the antibodies in KEY RESOURCES TABLE. Regarding the number of lasers, we also disagree with reviewer. When describing a specific model, it often happens that the same model has different number of lasers.

  1. For the data analysis method shown in Figure 4, add in the statistical data processing section either a reference or indicate which program was used (Visualisation with Circos) because it is not a program for classical statistical data processing. It is not mentioned anywhere in the material and method section. It is necessary to add the analysis to the material and method section.

Answer: The reference for Circos plot is cited “TCR repertoire data were visualized using VDJtools-1.2.1 (Shugay et al., 2015)"}" id="713240506">(Shugay et al., 2015)at ImmunoGeneTeqs, Inc (Chiba, Japan)” in TCR repertoire analysis in the Materials and Methods section (lines 529-530).

Round 3

Reviewer 2 Report

Comments and Suggestions for Authors

similar work was previously published by the authors

Comments on the Quality of English Language

ok

Author Response

Thank you for pointing to the previous manuscript. Indeed, we have deposited our recent manuscript in BioRiv on 4th January 2023 (Sugimoto et al., ID#: BIORXIV/2022/501791 TITLE: Mice generated with induced pluripotent stem cells derived from mucosal-associated invariant T cells).